# Efficacy of Antimicrobial Agents in Dentifrices: A Systematic Review

**DOI:** 10.3390/antibiotics11101413

**Published:** 2022-10-14

**Authors:** Vanessa Teixeira Marinho, Andréa Cândido dos Reis, Mariana Lima da Costa Valente

**Affiliations:** Department of Dental Materials and Prosthodontics, Ribeirão Preto Dental School, University of São Paulo (USP), Ribeirão Preto 14040-904, Brazil

**Keywords:** dentifrices, efficacy, antimicrobial activity, antimicrobial agents, in vitro

## Abstract

The aim of this systematic review was to verify if the presence of different antimicrobial agents in dentifrices is effective in reducing the number of microorganisms for disease prevention. This review followed the Preferred Reporting Items for Systematic Review and Meta-Analysis (PRISMA) guidelines and was registered with the Open Science Framework (OSF). A search was conducted in the PubMed, Embase, Scopus, and Web of Science databases. Two independent authors reviewed the titles and abstracts according to the inclusion criteria, which comprised in vitro studies published in English that evaluated the efficacy of antimicrobial agents in dentifrices and their antimicrobial activity. A total of 527 articles were found. Of these, 334 were included for reading of the title and abstract, and 69 were selected for reading in full. In the end, 39 articles remained in this review. Triclosan, sodium fluoride, and sodium monofluorophosphate were the most commonly used chemical antimicrobial agents. Among the herbal agents, miswak extract and neem extract were the most commonly used. The presence of antimicrobial agents in dentifrice formulations can promote the reduction of the number of microorganisms involved in oral diseases, but with variations in their effectiveness, depending on the agent used and the microorganism evaluated.

## 1. Introduction

The human oral cavity is a complex environment composed of a variety of microorganisms, including commensal bacteria that are part of the healthy oral microbiota and are important for physiology, and pathogenic microorganisms, which are responsible for various infections, the most common and prevalent of which are dental caries and periodontal diseases [1]. When the balance of an individual’s oral microbiota is lost, opportunistic pathogens can proliferate, resulting in the development of diseases [2].

Several microorganisms are involved in the progression of oral pathologies. *Streptococcus mutans* is a Gram-positive bacteria frequently found in the human oral cavity and one of the main microorganisms involved in the etiology of dental caries, along with *Lactobacillus* spp. [3,4,5]. *Candida albicans* is the most commonly commensal fungal species from the oral cavity, and under conditions of dysbiosis, can favor the appearance of oral candidiasis, an opportunistic infection, and is also associated with active caries lesions and promotes negative influences on tissue [6,7]. *Staphylococcus aureus* is another Gram-positive bacterium capable of causing opportunistic infections [8]. On the other hand, periodontal diseases are mainly caused by streptococci and spirochetes [6].

Caries is a multifactorial disease characterized by unbalanced mineralization and demineralization of teeth. *S. mutans* acts in the fermentation of carbohydrates from food, and this process produces acids that demineralize and degrade the dental element [6]. Biofilm accumulation and cariogenic microbiota are closely related to disease progression [9]. Periodontal diseases include gingivitis and periodontitis, which affect the tissue and structures that protect and support the teeth, respectively. Periodontitis, the most severe form of the disease, can result in loss of the dental element [6]. Biofilm accumulation also plays an important role in this infection [10].

Tooth brushing with dentifrice is one of the most frequently used oral hygiene practices in the world and is an essential measure for the maintenance of oral health [11,12,13]. Through it, it is possible to mechanically remove biofilm and consequently reduce the number of microorganisms [1]. However, it is not always properly performed. Therefore, the incorporation of antimicrobial agents in dentifrices is a prophylactic method of great importance to help control the number of microorganisms present in the oral cavity and thus reduce the chances of occurring infections. They act by slowing the microorganisms’ multiplication, preventing bacterial aggregation and rupture of the pathogens’ cell walls [14]. Antimicrobial agents such as sodium fluoride and triclosan have as their mechanism of action the inhibition of the activity of different enzymes [15,16]. In addition, herbal dentifrices contain phytochemicals, which are the substances responsible for their antimicrobial and anti-inflammatory effects [17].

Currently, there are several components present in dentifrice formulations with antimicrobial properties, such as triclosan and fluoride agents. However, some chemical antimicrobial agents can cause adverse effects [1,18]. Therefore, natural herbal dentifrices have emerged as a safer alternative to control and reduce the risk of infections and improve oral health. Herbal extracts have phytochemical components, which have antimicrobial and anti-inflammatory properties [17].

Most dentifrices are claimed to have antimicrobial properties, but comparative information on their efficacy is needed. There is a wide variety of dentifrices currently available, with various active substances, especially with the popularization of phytotherapeutic agents. Therefore, the aim of this systematic review was to verify if the presence of different antimicrobial agents in dentifrices is effective in reducing the number of microorganisms involved in oral diseases.

## 2. Materials and Methods

This systematic review was structured in accordance with Preferred Reporting Items for Systematic Review and Meta-Analysis (PRISMA) guidelines and was registered with the Open Science Framework (OSF). The question of the present review was formulated based on the PICOS: “Is the presence of antimicrobial agents in dentifrices really effective in reducing the number of microorganisms?” In this case, the population was formed by dentifrices without type restriction, the intervention was the presence of antimicrobial agents, the comparison was made with a group without the presence of antimicrobial agents, and the outcome was the evaluation of efficacy through antimicrobial activity, with the study focus being in vitro studies.

The electronic search for articles was conducted in November 2021 in the PubMed, Embase, Scopus, and Web of Science databases. The search was performed using the following terms: (dentifrices OR toothpastes) AND (“anti-infective agents” OR “anti-bacterial agents” OR “antimicrobial agents”) AND (“antibacterial activity” OR “antimicrobial activity”). The articles found were exported to Rayyan digital platform (Qatar Computing Research Institute, Doha, Qatar). Table 1 summarizes the database search strategy.

The initial selection was performed by two independent authors. In it, articles were selected based on titles and abstracts, and for studies with insufficient data, the manuscript was obtained and read in full. Studies that did not meet the established eligibility criteria were excluded. The second selection was made by reading the full text. Disagreements in the selection of articles were resolved through discussion between independent authors.

Eligibility criteria included in vitro studies published in English that evaluated the efficacy of antimicrobial agents in dentifrices and their antimicrobial activity. The exclusion criteria were applied to studies that did not evaluate dentifrices and studies that did not evaluate antimicrobial activity. Furthermore, in vivo studies, observational studies, studies that used clinical samples, reviews, book chapters, conference abstracts, case reports, surveys, and letters to the editor were also excluded.

Data from the included studies were extracted in a table in a Word document (Microsoft Corporation, Redmond, USA) with the following information: (1) authors and year of publication; (2) antimicrobial agents; (3) dentifrices used; (4) microorganisms evaluated; (5) method of antimicrobial activity evaluation; and (6) main conclusions (Table 1).

The methodological quality and risk of bias of the included studies were assessed by two authors by the Joanna Briggs Institute quasi-experimental study assessment tool, which was adapted for systematic review of in vitro studies. Each study was classified as low risk, uncertain risk, or high risk.

## 3. Results

Figure 1 depicts the study selection strategy. A total of 527 articles were identified in the initial search, of which 193 were duplicates. After analyzing the inclusion and exclusion criteria, 69 articles were selected for full-text reading. Of these, 30 were excluded, resulting in the selection of 39 relevant articles.

All selected articles evaluated the efficacy of different antimicrobial agents present in dentifrice formulations. Ten authors [9,19,20,21,22,23,24,25,26,27] evaluated dentifrices with chemical antimicrobial agents, nine authors [1,28,29,30,31,32,33,34,35] evaluated dentifrices with herbal antimicrobial agents and twenty authors [2,6,11,12,18,36,37,38,39,40,41,42,43,44,45,46,47,48,49,50] evaluated both chemical and herbal agents.

The main antimicrobial agents present in the formulations of the studied dentifrices were sodium fluoride [2,6,12,18,20,21,22,24,25,26,27,36,41,42,43,44,45,46,47,48] and triclosan [2,6,11,12,18,20,21,22,24,26,27,37,42,43,44,45,46,47]. The antimicrobial agent sodium monofluorophosphate was also present in several formulations [6,18,20,21,22,24,37,38,43,45,46,47,49]. Regarding herbal antimicrobial agents, miswak [6,32,37,44,45,46,47] and neem [37,44,45,46,47] extracts were the most commonly used. In addition, some authors [18,21,22,24] reported that sodium lauryl sulfate, usually added to dentifrices due to its detergent properties, is also capable of producing antimicrobial activity. In general, among the antimicrobial agents, the dentifrice formulations that have triclosan as the active agent showed the most significant antimicrobial effect. In the studies of Evans et al. [22], Anushree et al. [37], Ali et al. [12] and Sadeghi et al. [21], a larger mean diameter of inhibition zone of triclosan-containing dentifrice (44.6 mm, 40.67 mm, 38 mm and 20.4 mm, respectively) was observed compared to sodium fluoride-containing dentifrice (8.4 mm, 18 mm, 12.4 mm and 14.3 mm, respectively) against *S. mutans*.

Some authors [39,41,45,47,48] demonstrated that certain herbal dentifrice formulations can be as effective as dentifrice formulations with chemical antimicrobial agents. Furthermore, in five studies [6,32,37,46,47] which evaluated the antimicrobial action of dentifrices in different dilutions, it was observed that the diameter of the inhibition zones decreased with increasing dentifrice dilution.

The most investigated microorganisms were *Streptococcus mutans*, *Candida albicans* and *Staphylococcus aureus*, respectively. In general, most antimicrobial agents were able to reduce the amount of *S. mutans*, but with great variations in their effectiveness. The largest diameter of the inhibition zones for this microorganism were 5.12 mm, 22 mm, 30 mm and 44.6 mm in the studies of Thounaojam et al. [39], Leite et al. [38], Babu et al. [32] and Evans et al. [22], respectively. The same was observed for *S. aureus*, where the largest diameter of the inhibition zones were 7 mm, 16.5 mm, 29.67 mm and 49 mm in the studies of Shaheena et al. [28], Leite et al. [38], Anushree et al. [37] and Ali et al. [12], respectively. In five studies [6,29,35,38,48], *C. albicans* did not show zones of inhibition or showed greater resistance to growth inhibition by plant extracts compared to chemical agents. However, another five studies [2,37,41,43,45] showed better results for some herbal dentifrices compared to chemical antimicrobials.

Figure 2 and Figure 3 summarize the results of the quality evaluation of the studies. The risk of bias was evaluated with the use of an adapted quasi-experimental studies appraisal tool by the Joanna Briggs Institute. Of the 39 studies included in this systematic review, most had a low risk of bias. The exception was the criteria “Were outcomes measured in a reliable way?”, which showed a high risk of bias. The studies did not report whether the trials were performed by the same technician or the number of raters.

Another criterion that showed a higher risk of bias was “Was appropriate statistical analysis used?” This result is justified by the absence of the statistical methods used [19,20,23,24,25,26,30,31,32,33,35,40,45]. Finally, some studies [20,22,26,31,32,43,46] had a high risk of bias for the criterion “Was there a control group?” because they had no control group, but rather just compared groups. Because the studies showed heterogeneity in antimicrobial agents, dentifrices used and microorganisms evaluated, statistical analysis or meta-analysis was not possible.

## 4. Discussion

The addition of antimicrobial agents to oral care products has been suggested as an important strategy to assist in the control and reduction of microorganisms involved in various diseases, in order to improve oral health. This systematic review investigated the effectiveness of different antimicrobial agents in toothpaste formulations and found that the presence of these components can promote a real reduction in the number of microorganisms involved in oral diseases, but with varying levels of effectiveness.

Fluoride agents were the most frequently used, such as sodium fluoride (NaF) and sodium monofluorophosphate (MFP). Besides acting in the tooth remineralization process and helping to prevent dental caries, fluoride is capable of interfering with bacterial metabolism through the inhibition of enzymes, such as enolase [15]. However, fluoride agents at high levels can cause adverse effects. Excessive exposure to fluoride in children during the formation of permanent teeth can lead to enamel mineralization defects, characterizing dental fluorosis [51]. In a study by Haraszthy et al. [52], the analyzed fluoride-containing dentifrices showed different antimicrobial effects. In most of the studies included in this review that compared fluoride dentifrices, the diameter of their zones of inhibition also varied [6,12,18,21,22,37,41,43,45,46]. Fluoride is more active and has a greater capacity to interfere with microorganism proliferation under low pH conditions [22]. Since some culture media used have a neutral pH, such as Mueller–Hinton agar, the action of fluoride may have been affected. In in vivo studies, considering the pH conditions of the oral cavity, it is possible to observe a reduction in the plaque index when this antimicrobial agent is used. Binney et al. [53] performed a clinical trial that evaluated the plaque inhibitory properties of five dentifrices, all containing fluoride agents. There was a reduction in the participants’ plaque index, which ranged from 2.49 to 2.24 among the dentifrices over a 4-day period. Similarly, in a clinical study by Gupta et al. [54], participants used a fluoride dentifrice and achieved a plaque score reduction ranging from 1.86 to 1.28 over a 4-week period.

Triclosan is a non-ionic phenolic derivative with antimicrobial properties that has also been widely used. In a study by McMurry et al. [16], triclosan showed antimicrobial activity through inhibition of the enzyme enoyl-acyl reductase (ENR) transporter protein, which participates in the synthesis of fatty acids. It also has anti-inflammatory effects, as it acts in the inhibition of the cyclooxygenase/lipoxygenase pathways [55]. Furthermore, triclosan and fluoride are able to damage the bacterial inner membrane [12]. However, great attention has been paid to the possible long-term side effects that this antimicrobial agent can cause [18], including the increase in resistance of microorganisms [22,56]. Some commercial dentifrices have a combination of triclosan and copolymer of maleic acid and polyvinyl methyl ether (PVM/MA), in order to improve its retention in the oral cavity and its solubility in water, resulting in greater substantivity [57,58]. A commercial dentifrice formulation containing triclosan and sodium fluoride has been used in several studies as a positive control and has been shown to be effective in the formation of inhibition zones of various microorganisms [2,11,42,44]. In only three studies in this review [27,43,45] were other dentifrice formulations more effective than those containing triclosan. In the results of Benlatef et al. [43] and Gibraiel et al. [45], herbal dentifrice formulations were more effective than triclosan. Fernández et al. [27], on the other hand, demonstrated that dentifrices containing sodium fluoride and stannous fluoride were more effective. In nine other studies [6,12,18,20,21,22,24,37,46], dentifrices containing combinations of triclosan and sodium fluoride or sodium monofluorophosphate were more effective than the other compared dentifrices that did not have triclosan in their composition. Similarly to these findings, Haraszthy et al. [52] and Forbes et al. [59] reported that dentifrices containing triclosan as the active agent showed greater antimicrobial activity than the others, such as sodium fluoride. In a study by Haraszthy et al. [52], the minimum inhibitory concentration (MIC) of dentifrice containing only sodium fluoride as an antimicrobial agent was 30 µg/mL for *S. mutans*, 30 µg/mL for *S. aureus*, and 75 µg/mL for *C. albicans*. In contrast, the MIC of the dentifrice containing triclosan and sodium fluoride was 7.5 µg/mL for *S. mutans*, 15 µg/mL for *S. aureus*, and 30 µg/mL for *C. albicans*, showing better results. The minimum inhibitory concentration is the lowest concentration of a substance that inhibits microbial growth. In addition, in a study by Forbes et al. [59], the minimum bactericidal concentration (MBC) against anaerobic bacteria of dentifrice containing only sodium fluoride as antimicrobial agent was 3.3 to 12.5 mg/mL, while the MBC of dentifrice containing triclosan and sodium fluoride was 1.6 to 6.3 mg/mL, the latter being more effective at lower concentrations.

Phytotherapeutic antimicrobial agents can be an alternative to the use of chemical agents in children and adults, in view of the possible adverse effects they can cause. Several studies [2,18,32,37,48] have demonstrated that the antimicrobial activity of herbal dentifrices containing different extracts shows a wide variation. In this review, miswak (*Salvadora persica*) and neem (*Azadirachta indica*) extracts were the most present in natural dentifrice formulations. Miswak is obtained from the branches of the *Salvadora persica* tree and has been used as an oral hygiene device since antiquity [60]. In studies by Mohammed [60], Sivakumar et al. [61] and Adwan et al. [62], dentifrices containing miswak extract showed antimicrobial action against different microorganisms, which is in agreement with the studies evaluated in this review [32,37,44,45,47]. *Azadirachta indica*, known as neem, is a plant that belongs to the Meliaceae family found in Asia and Africa [63]. From it, a stick is made that is also used as an oral hygiene tool by some Asian and African countries [64]. In a study by Jenner et al. [65], the dentifrice containing neem extract was the most effective among those evaluated. Shafiq et al. [66] reported that dentifrice containing neem extract showed inhibitory activity. In the present review, some included studies [37,44,45,47] also demonstrated antimicrobial activity in dentifrices containing neem extract. Prasanth [6] and Anushree et al. [37] reported that dentifrice containing *Azadirachta indica* and *Salvadora persica* extracts formed halos of inhibition, but smaller in comparison to dentifrice containing triclosan. It is important to emphasize that most herbal dentifrices do not contain only one extract, but a combination of them. Thus, the variations in efficacy between herbal dentifrices probably occurs because of the difference between the components present in each brand.

Most of the assays were performed using the agar diffusion test, used to compare the antimicrobial activity of the products. In this method, the physicochemical properties of each antimicrobial agent, such as the diffusion coefficient and solubility, have an influence on its diffusion through the agar matrix and, consequently, on the results obtained [14,18]. However, this is a good option as a screening before performing in vivo studies. The test can be done using paper disks containing the agent or in drilled wells, where the agents are introduced. The disk diffusion method is appropriate for fluid products, but it has been widely used to evaluate the antimicrobial activity of dentifrices, which are semi-solid, but become fluid upon contact with water or saliva [67].

Another point to be considered is the interaction between the different components of dentifrices. The presence of some substances in the formulation of dentifrices can either increase or decrease the substantivity and clinical activity of some component [68]. In a study by Sadeghi and Assar [21], two commercial dentifrices had triclosan and sodium monofluorophosphate as antimicrobial agents, but they differed in other components. Although the antimicrobials used were the same, the antimicrobial activity of the dentifrices was different. Thus, the synergistic interactions between the various ingredients present in the formulations are an important factor not to affect the effectiveness of the product [38]. Sadeghi and Assar [21] observed that the antimicrobial action against the microorganisms evaluated was greater in dentifrices that contained more than one antimicrobial agent in their formulation. Following the same reasoning, in a study by Ali et al. [12], commercial dentifrices that already contained antimicrobial agents showed larger zones of inhibition after the addition of Piper betle extract. Oluwasina et al. [41] reported that dentifrice formulations containing a mixture of three or two extracts showed greater bioactivity. In this study, the dentifrice containing three extracts, *Dennettia tripetala*, *Syzygium aromaticum*, and *Jatropha curcas* latex showed inhibition zones against several microorganisms ranging from 10 to 18.3 mm, while the dentifrice containing two extracts, *Syzygium aromaticum* and *Jatropha curcas* latex, showed zones ranging from 9 to 16 mm. In contrast, the dentifrice containing only one extract, *Jatropha curcas* latex, exhibited inhibition zones ranging from 0 to 10 mm. Also, the dentifrice containing only *Dennettia tripetala* showed zones of inhibition ranging from 0 to 8 mm, demonstrating the greater efficacy of dentifrices containing three and two extracts.

In addition, five authors [6,32,37,46,47] evaluated dentifrices in different dilutions and reported that the diameter of the inhibition zones decreased with increasing dilution of the dentifrice, which suggests that the efficacy of the antimicrobial agent may be reduced when it is diluted. The evaluation of dentifrice at various concentrations is of interest, since under in vivo conditions, saliva is able to dissolve the product.

## 5. Conclusions

Based on the results of this systematic review, some conclusions were established.

The presence of antimicrobial agents in dentifrice formulations can promote the reduction of the number of microorganisms involved in oral diseases, but with variations in their effectiveness, depending on the agent used and the microorganism evaluated.Some dentifrice formulations with herbal ingredients, such as miswak and neem extracts, can be as effective as dentifrice formulations with chemical antimicrobial agents, such as sodium monofluorophosphate and sodium fluoride.The antimicrobial activity of a dentifrice with antimicrobial agents can be reduced when it is diluted.The interaction between the different components of a dentifrice can influence the effectiveness of its antimicrobial activity, and thus the synergism between the ingredients is of great importance.

## Figures and Tables

**Figure 1 antibiotics-11-01413-f001:**
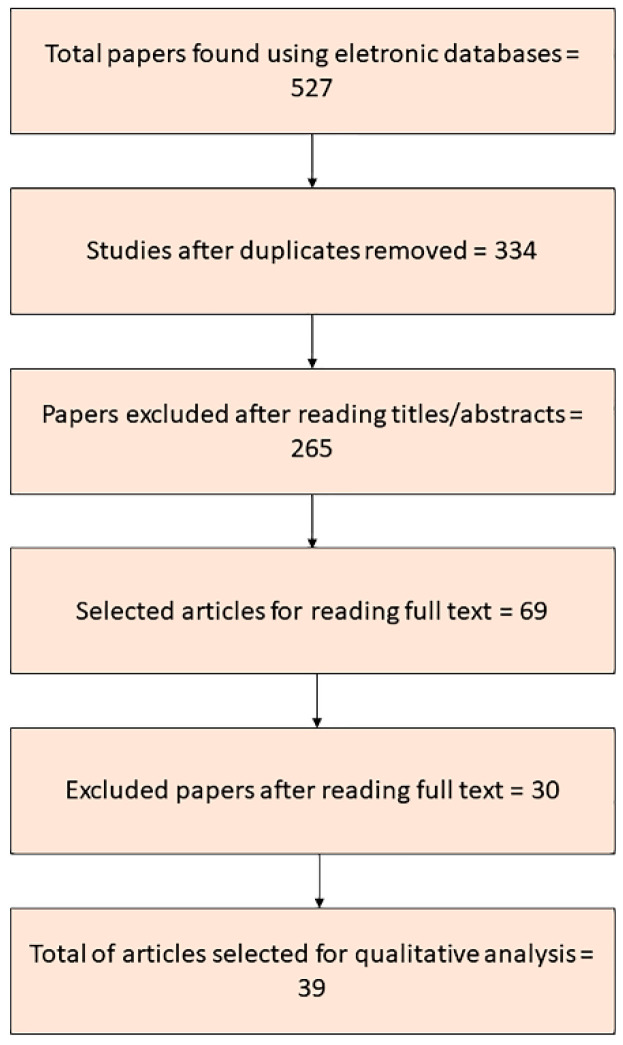
Flowchart summarizing the phases of the systematic review.

**Figure 2 antibiotics-11-01413-f002:**
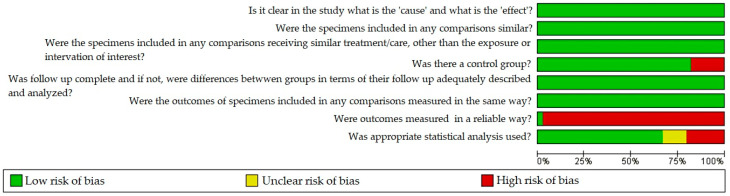
Risk-of-bias graph of included studies.

**Figure 3 antibiotics-11-01413-f003:**
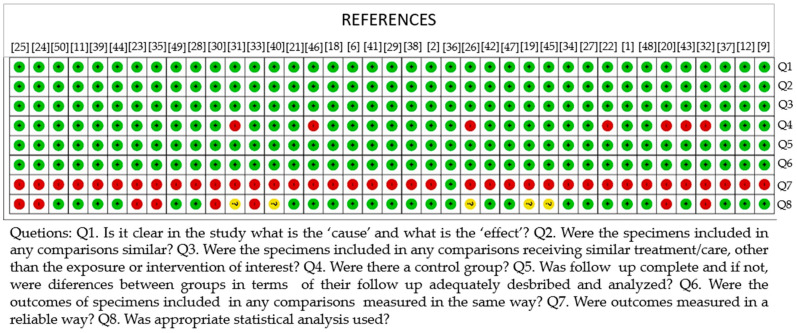
Risk-of-bias summary of included studies.

**Table 1 antibiotics-11-01413-t001:** Summary of included studies.

Material/Antimicrobial Agent	Dentifrices Used	Organisms	Method of Antimicrobial Activity Evaluation	Main Conclusions	Ref.
Cinnamon, rosemary, nutmeg, orange, mint, ginger, oregano, thyme, clove, eucalyptus, tangerine, lime and tea tree oil	18 dentifrices formulated with essential oils of cinnamon, clove, oregano and thyme	*Staphylococcus aureus*, *Streptococcus mutans*, *Lactobacillus lactis* and *Enterococcus faecalis*	Hole-plate diffusion method Broth microdilution method (MIC)	In general, the best results were obtained with cinnamon essential oil and dentifrice against *S. mutans*	[1]
Triclosan, bee propolis and plant extracts, such as Peelu, neem, licorice, pomegranate rind, clove, Persian walnut, cinnamon, peppermint, echinacea, grapefruit seed, among others	14 dentifrices: Acu-Herb, Auromere Herbal Toothpaste, Dental Gel, Dental Herb (powder), Dentifrico de Echinacea, Healthy Mouth, Herbal Brite, Nature’s Gate Natural Toothpaste, Nutrismile C, Peelu, Pink Toothpaste with Myrrh, Pure Herb, Sea Fresh and Tom’s of Maine Natural Toothpaste	*Streptococcus mutans*, *Streptococcus sanguis*, *Actinomyces viscosus* and *Candida albicans*	Standard diffusion method	6 herbal dentifrices were effective in inhibiting the growth of microorganisms Only one herbal dentifrice showed consistent antimicrobial activity against all 4 microorganisms (Dental Gel)	[2]
Triclosan, sodium monofluorophosphate, sodium fluoride and herbal products, such as *Salvadora persica*, *Azadirachta indica*, Babhul, among others	5 dentifrices (A, B, C, D and E) and 5 mouthwashes	*Streptococcus mutans*, *Escherichia coli* and *Candida albicans*	Modified agar well diffusion method	Triclosan containing dentifrice formulations were more effective compared to non-triclosan containing synthetic dentifrices Herbal dentifrices exhibited lower efficacy compared to the others	[6]
Nanosilver, chitosan and fluoride	3 dentifrices: containing nanosilver (TruCare Nanosilver); containing chitosan (Conybio Plus Chitosan); and containing fluoride (Oral B Pro-Health)	*Streptococcus mutans*	Modified agar well diffusion method	The dentifrice containing nanosilver has the highest antibacterial efficacy against *S. mutans*, followed by the dentifrice containing fluoride and chitosan	[9]
Rosemary extract, propolis, mauve, cinnamon, peppermint, triclosan and sodium fluoride	Dentifrices based on rosemary (TR) and propolis (TH—Sorriso Herbal com Própolis) and Colgate Total 12 (TPC)	*Streptococcus mutans*, *Streptococcus oralis* and *Lactobacillus rhamnosus*	Macrodilution method Standardized single disk method	The antimicrobial activity of rosemary-based dentifrice (TR) was similar to the commercial dentifrice (TH) For the inhibition of *L. rhamnosus*, the propolis-based commercial dentifrice was more effective than TR dentifrice	[11]
*P. betle* extract, triclosan, sodium fluoride, sodium bicarbonate, corn mint, peppermint, sage oil and coneflower	3 dentifrices: Colgate Total, Paradontax and Darlie	*Escherichia coli*, *Staphylococcus aureus*, *Streptococcus mutans*, *Streptococcus salivarius* and *Candida albicans*	Standard procedure for determining MIC Agar disk diffusion assay	A statistically significant increase in the zone of inhibition after addition of *P. betle* extract was observed with all 3 dentifrices on all pathogens tested	[12]
Triclosan, sodium monofluorophosphate, sodium fluoride, sodium lauryl sulfate and plant extracts, such as peppermint, eucalyptus, cinnamon, clove, Carum petroselinum (parsley), among others	15 dentifrices and a stannous fluoride gel: Sensodyne Repair, Sensodyne Total Care, Pronamel, Colgate Pro Relief, Colgate Sensitive Multi, Colgate Total, Colgate Neutrafluor 5000 plus, Colgate Sparkling Gel, Alfree, Macleans Advanced Enamel Lock, Gel Kam, Herbal Brite, Herbal Fresh, Grants and Woolworthse Select	*Streptococcus mutans*	Agar diffusion method	Colgate Total has the greatest effect and Colgate Gel Kam has the least. Only two herbal toothpastes showed antimicrobial activity (Herbal Fresh and Macro) Most of the antimicrobial activity against *S. mutans* depends on the presence of non-fluoride agents, such as triclosan and sodium lauryl sulfate	[18]
Silver and gold nanoparticles	Royal Denta dentifrice containing silver and gold nanoparticles	*Staphylococcus aureus*, *Enterococcus faecalis*, *Escherichia coli*, *Klebsiella pneumoniae*, *Proteus mirabilis*, *Pseudomonas aeruginosa*, *Bacillus cereus* and *Candida albicans*	Agar diffusion method	Silver in dentifrice has a greater antimicrobial effect than gold, but its effect is still lower than that of a chemical antimicrobial agent	[19]
Sodium fluoride, sodium monofluorophosphate and triclosan	3 desensitizing dentifrices (Colgate Sensitive Pró-Alívio, Sensodyne Rápido Alívio and Oral B Pro Sensitive) and a common dentifrice (Colgate Total 12 Professional Clean)	*Candida albicans*, *Streptococcus mutans* and *Staphylococcus aureus*	Microdilution method (MIC)	Colgate dentifrice was able to eliminate all microorganisms evaluated at lower concentrations compared to Colgate Sensitive and Oral B Sensitive *C. albicans* was the least resistant to the dentifrices. *S. aureus* was the most resistant.	[20]
Sodium monofluorophosphate, sodium lauryl sulfate, triclosan, bromochlorophene and sodium fluoride	10 Iranian-made dentifrices: Paveh, Saviz, Latifeh II, Bath, Darugar II, Darugar I, Close up, Tage, Pooneh III and Nasim	*Streptococcus mutans*, *Streptococcus sanguis*, *Actinomyces viscosus* and *Candida albicans*	Agar diffusion method	All dentifrices showed antimicrobial activity. Those containing more than one antimicrobial showed higher activity The antimicrobial property of Pooneh III was similar and Bath higher than the positive control	[21]
Sodium monofluorophosphate, stannous fluoride and sodium fluoride in various concentrations	Dentifrices with low fluoride concentration: Colgate My First, Oral B Stages and Macleans Milk Teeth Dentifrices with standard fluoride concentration for adults: Colgate Total, Oral B Tooth and Gums, Macleans Extreme Clean, Colgate Sensitive Pro Relief and Oral B Pro Health Dentifrice with high fluoride concentration: Colgate Neutrafluor 5000	*Streptococcus mutans*, *Streptococcus sanguinis* and *Lactobacillus acidophilus*	Agar diffusion assay	Dentifrices containing 1450 ppm fluoride produced greater growth inhibition of *S. mutans* and *S. sanguinis* than those with <500 ppm Colgate Total produced the highest average growth inhibition on *S. mutans* and *S. sanguinis*. Colgate Pro Relief produced the lowest	[22]
Stabilized chlorine dioxide	ClōSYS oral care products: two mouth rinses, one oral spray and one fluoride dentifrice	SARS-CoV-2, SARS-CoV, human coronavirus (HCoV) 229E, influenza A (H3N2), rhinovirus type 14, adenovirus type 5 and herpes simplex virus (HSV) type 1 and 2	In vitro suspension virucidal assays	Stabilized chlorine dioxide contained in oral care products reduced the viral load of several viruses	[23]
Sodium monofluorophosphate, sodium fluoride, triclosan and zinc citrate	5 dentifrices: Colgate Regular, Macleans Anti-plaque Formula, Mentadent P Gum Health Formula, Crest Gum Health and Colgate Gum Protection	17 bacterias: *A. actinomy-cetemcomitans* (2), *A. odontolyticus*, *A. viscosus*, *C. rectus*, *C. orchracea*, *Capnocytophaga species*, *E. timidum*, *S. oralis*, *P. anaerobius*, *P. micros*, *P. gingivalis*, *P. intermedia* (2), *S. constellatus*, *V. parvula* and *Veillonella species*	Agar dilution method Maximum inhibitory dilution (MID)	Mentadent Gum Health Formula, which contains triclosan and zinc citrate, had the highest antimicrobial activity of the dentifrices containing triclosan Crest Gum Health was generally less active than all other dentifrices, including Colgate Regular, except against *A. actinomycetemcomitans*	[24]
Stannous fluoride, sodium fluoride and stannous pyrophosphate	2 stannous fluoride dentifrices (SF1, SF2), 2 experimental dentifrices with SF and stannous pyrophosphate (SFSP1, SFSP2), a gel with SF (G) and a dentifrice with NaF (C)	20 species: *Actinobacillus actinomycetemcomitans* (2), *Campylobacter rectus*, *Capynocytophaga sputigena*, *Capynocytophaga species*, *Fusobacterium nucleatum*, *Porphyromonas gingivalis* (2), *Prevotella nigrescens*, *Prevotella intermedia*, *Veillonella parvala*, *Veillonella species*, *Actinomyces viscosus*, *Actinomyces odontolyricus*, *Eubacterium timidum*, *Peptostreptococcus anaerobius*, *Peptostreptococcus micros*, *Streptococcus constellatus*, *Streptococcus mutans* and *Streptococcus oralis*	Maximum inhibitory dilution (MID)	All formulations showed antimicrobial activity with the average order of highest activity descending being C, SF2, SF1, SFSP1, SFSP2 and G SF1 containing stannous fluoride was less active than C, a conventional dentifrice containing sodium fluoride (NaF)	[25]
Triclosan and fluoride	Colgate Total dentifrice	*Klebsiella pneumoniae*, *Streptococcus mutans*, *Candida albicans*, *Porphyromonas gingivalis* and *Fusobacterium nucleatum*	Microbial kill time assay European suspension test method	In antimicrobial assays, the dentifrice was effective against bacteria, but not against *C. albicans*	[26]
Sodium fluoride, triclosan and stannous fluoride	4 commercially available dentifrices, 2 containing sodium fluoride (NaF) at different concentrations (1450 and 2500 ppm) and 2 NaF with triclosan or stannous fluoride	*Streptococcus oralis*, *Actinomyces naeslundii*, *Veillonella parvula*, *Fusobacterium nucleatum*, *Porphyromonas gingivalis* and *Aggregatibacter actinomycetemcomitans*	Subgingival biofilm model	Dentifrices containing NaF and stannous fluoride demonstrated higher antimicrobial activity for *A. actinomycetencomitans*, *P. gingivalis*, and *F. nucleatum* when compared to those containing NaF and triclosan, 1450 ppm NaF, or 2500 ppm NaF	[27]
Guava leaf extract	Three prepared toothpaste formulations varying in ingredient concentrations (F1, F2 and F3)	*Bacillus subtilis*, *Proteus vulgaris*, *Staphylococcus aureus*, *Streptococcus mutantes* and *Streptococcus oralis*	Well diffusion method	F3 was very effective against the bacteria tested, followed by F2 and F1 The highest zone of inhibition was observed against *P. vulgaris* and *B. subtilis* and the lowest was observed against *S. aureus*	[28]
*Solanum tuberosum* powder	Experimental dentifrice containing *Solanum tuberosum* (Tocosh)	*Staphylococcus aureus*, *Streptococcus mutans*, *Streptococcus mitis* and *Candida albicans*	Agar diffusion method	Dentifrice with Tocosh has an antibacterial effect against *S. aureus*, *S. mutans* and *S. mitis*. No antimicrobial effect was found against *C. albicans*	[29]
Pomegranate peel, lemon peel and mango peel extract, mint oil and sodium lauryl sulfate	A polyherbal dentifrice containing pomegranate, lemon and mango peel methanolic extract formulated in different concentrations (100, 250 and 500 mg/mL)	*Staphylococcus aureus*, *Bacillus cereus*, *Escherichia coli* and *Pseudomonas aeruginosa*	Disk-diffusion method	The polyherbal dentifrice has promising antimicrobial effects against Gram-positive and Gram-negative organisms, and may be safer compared to a synthetic dentifrice *S. aureus* was the most sensitive, followed by *E. coli*, *B. cereus* and *P. aeruginosa*	[30]
Propolis (10% ethanol solution)	Yamada Yohojo Propolis Hamigaki dentifrice, which contains an ethanolic solution of 10% propolis sample at a concentration of 2% (*w*/*w*)	20 strains representing 12 species: *S. mutans* (3), *S. sobrinus* (3), *S. cricetus*, *S. rattus*, *S. sanguis* (2), *S. oralis*, *S. gordonii*, *S. mitis* (2), *S. salivarius*, *A. naeslundii*, *A. viscosus* (3) and *L. casei*	Minimum inhibitory concentration (MIC)	The propolis-containing dentifrice inhibited the growth of 20 bacterial strains in a concentration range of 3–7 mg/mL	[31]
Ingredients not specified	6 dentifrices: Ayush, Dant kanti, Colgate vedshakti, Meswak, Dantajeevan and Dabur red	*Streptococcus mutans* and *Candida albicans*	Antibiotic susceptibility testing Agar disk-diffusion method	The herbal dentifrice formulations were effective in controlling the oral microflora For *S. mutans*, the zone of inhibition is greatest in Ayush. For *C. albicans*, the zone of inhibition is greatest in Ayush and Dant kanti	[32]
Green tea ethanolic extract and sodium lauryl sulfate	Two experimental dentifrice formulas containing 5% green tea extract in a gel base. Formula A was enriched with 30% CaCO_3_ as an abrasive. Formula B was made without CaCO_3_	*Streptococcus mutans* and *Lactobacillus acidophilus*	Agar diffusion method	In comparison with herbal dentifrice patent containing a combination of green tea extract and *P. bettle* leaves extract, this had higher antibacterial activity against *S. mutans* than the green tea dentifrice, but not against *L. acidophilus*	[33]
*G. kola* extract, sodium lauryl sulfate and sodium bicarbonate Close up, Aquafresh and Colgate do not have their ingredients specified	Experimental dentifrice containing *G. kola* extract, Close up, Aquafresh and Colgate	*Escherichia coli*, *Pseudomonas aeruginosa*, *Proteus mirabilis*, *Klebsiella pneumoniae*, *Bacillus subtilis*, *Staphylococcus aureus*, *α-haemolytic*, *Streptococcus pneumoniae*, *Streptococcus pyogenes* and *Candida albicans*	Standard agar diffusion and broth dilution methods	All microorganisms were significantly susceptible to the extract and the dentifrice The herbal dentifrice inhibited all tested microorganisms, while none of the commercial dentifrices used showed any antimicrobial activity	[34]
Herbal extracts, such as *Stevia rebaudiana* leaves (SR), *Azadirachta indica* bark (AZ), *Piper longum* fruit (PL), *Curcuma longa* rhizome (CL), *Salvadora persica* bark (SP), among others	Polyherbal dentifrice (formulations 1, 2 and 3), Herbodent and Apollo	*Streptococcus mutans*, *Streptococcus oralis*, *staphylococcus aureus*, *Candida albicans* and *Lactobacillus acidophilus*	Agar well diffusion method Serial micro dilution method (MIC)	The methanol extract of polyherbal formulation 1 and 3 showed lower activity when compared to polyherbal formulation 2. However, the activity was lower than gentamicin	[35]
Calendula extracts, sage clay, sodium fluoride and sodium lauryl sulfate	2 dentifrices: Iranian herbal dentifrice and Iranian chemical dentifrice	*Streptococcus mutans*, *Lactobacillus* and *Candida albicans*	Agar disk-diffusion method	At the total concentration, herbal and chemical dentifrices have the same antimicrobial effect, but when reducing the concentration, the effect of the herbal dentifrice is reduced compared to the chemical one	[36]
Triclosan, sodium monofluorophosphate and plant extracts, such as miswak, neem, Babool, Pudina, Long, Vajradanti, turmeric, calendula, eucalyptus, and others	9 dentifrices in three groups	*Escherichia coli*, *Staphylococcus aureus*, *Streptococcus mutans* and *Candida albicans*	Agar well-diffusion method	Of the herbal groups, the only dentifrice containing several phytochemicals was found to be significantly effective and comparable to the triclosan and fluoride formulation. Homeopathic products showed lower antimicrobial activity	[37]
*Ricinus communis*, sodium, monofluorophosphate, chloramine T and sodium bicarbonate	Experimental dentifrice based on *Ricinus communis* Colgate, Corega Brite and Trihydral commercial toothpastes	*Staphylococcus aureus*, *Escherichia coli*, *Streptococcus mutans*, *Enterococcus faecalis*, *Candida albicans* and *Candida glabrata*	Microdilution technique in 96-well plates (MIC) Well agar diffusion method	Comparing the experimental dentifrices, the product with 10% *R. communis* produced the largest inhibition halos and showed antimicrobial activity similar to commercial dentifrices, except against *S. aureus* None of the dentifrices were effective against *E. coli*	[38]
Fluoride and herbal extracts	Herbal dentifrices: Himalaya herbals (A1) and Dabur red (A2) Conventional dentifrices: Colgate super shakti (B1) and Pepsodent complete germicheck (B2)	*Streptococcus mutans* and *Candida albicans*	Disk-diffusion method	Herbal dentifrices are equally and sometimes better than conventional ones At 50% concentration, B2 showed the maximum zone of inhibition for *S. mutans*, while at 100% concentration A1 showed better effects. For *C. albicans*, A2 was the most effective	[39]
Nanohydroxyapatite (nanoHAP) and *Curcuma aeruginosa* extract	20 dentifrices with different concentrations of nanoHAP and *C. aeruginosa*, plus OF1 and OF2	*Streptococcus mutans*	Agar diffusion and microdilution methods	Most of the dentifrice formulations showed antibacterial activity. OF1 and OF2 were shown to have antibacterial activity comparable to that achieved with the control	[40]
Ethanolic extracts of *Dennettia tripetala* seeds, *Syzygium aromaticum* buds, *Jatropha curcas* latex, sodium fluoride and *Aloe vera*	3 commercial dentifrices labeled Com A, Com B and Com C The formulated dentifrices were labeled DenSyzJat, DenSyz and SyzJat	*Escherichia coli*, *Bacillus* sp., *Staphylococcus aureus*, *Staphylococcus aureus* resistente à meticilina, *Staphylococcus epidermidis*, *Micrococcus luteus*, *Streptococcus mutans*, *Streptococcus pyogenes*, *Lactobacillus acidophilus*, *Candida albicans* and *Pseudomonas aeruginosa*	Agar well diffusion method Minimum inhibitory concentration (MIC)	The formulated dentifrices exhibited antimicrobial property against all tested microorganisms and showed better and significant antimicrobial effect when compared to commercial dentifrices The dentifrice formulated with *S. aromaticum* extract alone seems to be much more active among the 3 bioactive materials used for formulation	[41]
*Origanum dubium* and *Cinnamomum cassia* oils, triclosan, sodium fluoride, sodium bicarbonate and plant extracts, such as *Calendula officinalis*, *Aloe barbadensis* and *Melaleuca alternifolia* oil	4 dentifrices: Splat Organic, Splat Biocalcium, Jack N ‘Jill and Colgate Total	*Streptococcus mutans*	Agar diffusion method in disk (pure oil) and well (dentifrice)	The antibacterial activity of the dentifrices was higher than positive control Herbal dentifrices showed higher antibacterial activity than their initial forms after the addition of essential oils. *C. cassia* showed higher antibacterial activity than *O. dubium*	[42]
Sodium monofluorophosphate, sodium fluoride, triclosan, sodium lauryl sulfate, herbal extracts and oils, such as *Clinacanthus nutans*, *Streblus asper* and *Murraya paniculata* leaves, peppermint oil, eucalyptus oil, among others	6 dentifrices: Darly, Close up, Systema, Herbal Twin Lotus, Salz and Colgate	*Candida albicans*	Agar diffusion assay Broth microdilution technique (MIC and MFC)	All dentifrices inhibited the growth of *C. albicans* with a mean range of the zone of inhibition between 8 and 16.92 mm Herbal Twin Lotus showed the largest mean zone of inhibition. Saltz showed the lowest mean zone of inhibition	[43]
Triclosan, sodium fluoride and herbal extracts such as neem, chamomile, Babul and Miswalk	6 dentifrices: neem, Vicco Vajradanti, Himalaya Herbal, Colgate Herbal, Dabur Red and Dabur Babool	*Streptococcus mutans* and *Lactobacillus acidophilus*	well method of microbial culture	Herbal dentifrices are more effective against lactobacilli organisms at the same level as conventional dentifrice and less effective for *S. mutans* when compared to non-herbal dentifrices	[44]
Triclosan, sodium monofluorophosphate, sodium bicarbonate, sodium lauryl sulfate, sodium fluoride and plant extracts, such as *Anacyclus pyrethrum*, *Azadirachta indica* (neem), *Acacia arabica* (Babool), *Salvadora persica* (miswak), Pudina, among others	7 dentifrices (A, B, C, D, E, F, and G) and 2 mouthwashes (H and I)	*Escherichia coli* and *Candida albicans*	Modified agar well diffusion method	The formulations containing natural antimicrobials were more effective compared to dentifrices containing synthetic antimicrobials The antibacterial activity of A is lower compared to formulation B at higher dilutions	[45]
Sodium monofluorophosphate, triclosan, zinc sulfate, sodium lauryl sulfate, amine fluoride, sodium fluoride and plant extracts, such as neem and miswak	7 commercial dentifrices	*Streptococcus mutans*, *Escherichia coli* and *Candida albicans*	Well agar diffusion assay	Dentifrice formulations containing triclosan are more effective The zone of inhibition decreases with increasing dilution	[46]
Sodium fluoride, triclosan, sodium monofluorophosphate, sodium lauryl sulfate, eugenol, clove oil and plant extracts, such as miswak and neem,	6 dentifrices: 3 fluoride (Colgate Total, Fresh and White and Safi–clove) and 3 3 non-fluoride dentifrices (Mukmin, Halagel and Pureen)	*Streptococcus mutans*	Standard disk diffusion method	All dentifrices showed antibacterial activity at both concentrations tested, but better results at full strength compared to the diluted. The antibacterial activity of the non-fluoridated toothpastes is as good as the fluoridated ones	[47]
Cashew extract, mango extract, calendula extract (*Callendula officinalis*), *Aloe vera*, natural banana, natural apple, sodium fluoride and sodium lauryl sulfate	6 infant dentifrices: experimental cashew-based dentifrice; experimental mango-based dentifrice; experimental dentifrice without plant extract and fluoride; First Teeth dentifrice; Weleda dentifrice; and Tandy dentifrice	*Streptococcus mutans*, *Streptococcus sobrinus*, *Lactobacillus acidophilus* and *Candida albicans*	Agar plate diffusion test	First Teeth, Weleda, mango-based dentifrice and dentifrice without plant extract showed no antimicrobial effect against any of the microorganisms tested The cashew dentifrice showed significant antimicrobial activity against *S. mutans*, *S. sobrinus* and *L. acidophilus*. Tandy had antimicrobial activity against all microorganisms	[48]
Sodium monofluorophosphate, sodium lauryl sulfate, and herbal extracts, such as maricha, pippali, sunthi, kapor, akarkara, khadir, lawang, among others	Marketed allopathic powdered toothpastes (brand I and II) and herbal powdered toothpastes (brand III and IV)	*Staphylococcus sorbinus*, *Staphylococcus salivarius* and *Lactobacillus acidophilus*	Agar well diffusion method	The diameter of the zone of inhibition observed in brands I and II for test microorganisms was not significantly different from the diameter observed in brands III and IV	[49]
Propolis and chlorhexidine	3 dentifrices (with propolis, without propolis and with chlorhexidine) and 2 mouthwashes (with propolis and chlorhexidine)	*Actinomyces naeslundii*, *Veillonella dispar*, *Fusobacterium nucleatum*, *Streptococcus mutans*, *Streptococcus oralis* and *Candida albicans*	In vitro multispecies biofilm model	Propolis seems to have no effect in respect to reducing CFU in the supragingival biofilm model used, neither in the dentifrice nor in the mouthwash	[50]

## Data Availability

Not applicable.

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
