# Peer review of "Efficacy of Antimicrobial Agents in Dentifrices: A Systematic Review"

_antibiotics, 2022, doi:10.3390/antibiotics11101413_

Round 1
Reviewer 1 Report
The reviewer really appreciates the efforts of the authors to conduct this study which has a good clinical significance. However, there several errors in the manuscript that needs to be revised to improve the quality of the manuscript. The reviewer would like to suggest the following revision in the manuscript to make it suitable for publication.
The author should follow the correct sequence in the manuscript as per the journal's instruction. Eg introduction- materials & methods- result- discussion-conclusions
Figure 2 and figure 3 need to be formatted. The font is too small to read. Please use appropriate font size and orientation.
Table 2 needs to be formatted by removing irrelevant information rather than copy paste the entire conclusion of the study. The table can be split as per the materials/ microbes/ time etc. for better understanding. It is impossible for any reader to correlate the information in one table extending 24 pages.
Reviewer 2 Report
Dear Authors, this is an interesting manuscript, only there are a few things that would need improving before it is suitable for publishing. The first (and the main issue) - is English - it is fine, only there are some points at which it needs stylistic improvement (such as e.g. Miswak extract and Neem extract were the most used - it should say: were "the most commonly/frequently etc. used". However, this can easily be improved. When it comes to other things I would suggest:
Candida albicans is the most common fungal species isolated from 39 the oral cavity and is also associated with active caries lesions [6], in addition to oral can-40 didiasis, an opportunistic infection. – some authors believe candida is a commensal and exerts a negative influence in the tissues in a certain environment – please review that
They act by inhibiting enzymatic activity and cause 58 the disruption of the cell walls of pathogens [13]. – please provide more details on what exactly different compounds do
90-91 In general, among the antimicrobial agents, the formulations containing 90 triclosan showed the most significant antimicrobial effect. – provide exact numbers/details
98-100 In general, most antimicrobial agents were 98 able to reduce the amount of S. mutans, but with great variations in their effectiveness. 99 The same was observed for the S. aureus microorganism. In five studies – provide exact numbers/details
138-140 Fluoride is more active and has a greater capacity 138 to interfere with microorganism proliferation under low pH conditions [19]. Since some 139 culture media used have a neutral pH, such as Mueller Hinton agar, the action of fluoride 140 may have been affected – please try to provide a comparison with in-vivo conditions – how can/ or can the results be translated into clinical practice?
158-164 – this sounds more like results. Try to compare and contrast triclosan containing toothpastes with other
208-209 Oluwasina et al. [38] reported that 208 dentifrice formulations containing a mixture of three or two extracts showed greater bio-209 activity. – exact numbers please/a little bit more detailed description
Conclusions
Some herbal dentifrice formulations can be as effective as dentifrice formulations with chemical antimicrobial agents. – please provide what herbal formulations (e.g. “Dentifrice formulations with herbal ingredients such as .xxxxx, )
Also, I believe Table 1 is not crucial for understanding the study and can be left out.
Kind regards!
Round 2
Reviewer 1 Report
No additional comments